# Antiproliferative Effect of Colonic Fermented Phenolic Compounds from Jaboticaba (*Myrciaria trunciflora*) Fruit Peel in a 3D Cell Model of Colorectal Cancer

**DOI:** 10.3390/molecules26154469

**Published:** 2021-07-24

**Authors:** Paula Rossini Augusti, Andréia Quatrin, Renius Mello, Vivian Caetano Bochi, Eliseu Rodrigues, Inês D. Prazeres, Ana Catarina Macedo, Sheila Cristina Oliveira-Alves, Tatiana Emanuelli, Maria Rosário Bronze, Ana Teresa Serra

**Affiliations:** 1Departamento de Ciências dos Alimentos, Instituto de Ciência e Tecnologia de Alimentos (ICTA), Universidade Federal do Rio Grande do Sul (UFRGS), Porto Alegre 91501-970, Brazil; eliseu.rodrigues@ufrgs.br; 2Núcleo Integrado de Desenvolvimento em Análises Laboratoriais (NIDAL), Centro de Ciências Rurais, Departamento de Tecnologia e Ciência dos Alimentos, Universidade Federal de Santa Maria, Santa Maria 97105-900, Brazil; deiaquatrin@hotmail.com (A.Q.); reniusmello@gmail.com (R.M.); tatiana.emanuelli@ufsm.br (T.E.); 3Departamento de Nutrição, Universidade Federal de Ciências da Saúde de Porto Alegre (UFCSPA), Porto Alegre 91501-970, Brazil; vivian_bochi@yahoo.com.br; 4iBET, Instituto de Biologia Experimental e Tecnológica, Apartado 12, 2780-901 Oeiras, Portugal; ines.duarte.96@gmail.com (I.D.P.); catarina.macedo@ibet.pt (A.C.M.); sheilacris.oliveira.alves@gmail.com (S.C.O.-A.); mbronze@ibet.pt (M.R.B.); tserra@ibet.pt (A.T.S.); 5Instituto de Tecnologia Química e Biológica António Xavier, Universidade Nova de Lisboa (ITQB NOVA), 2780-157 Oeiras, Portugal; 6iMed, Faculdade de Farmácia da Universidade de Lisboa, Av das Forças Armadas, 1649-019 Lisboa, Portugal

**Keywords:** spheroids, HT29 cells, cluster analysis, principal component analysis, hydrolysable tannins, HHDP-digalloylglucose isomer, dihydroxyphenyl-*γ*-valerolactone

## Abstract

Jaboticaba is a Brazilian native berry described as a rich source of phenolic compounds (PC) with health promoting effects. PC from jaboticaba peel powder (JPP) have low intestinal bio-accessibility and are catabolized by gut microbiota. However, the biological implication of PC-derived metabolites produced during JPP digestion remains unclear. This study aimed to evaluate the antiproliferative effects of colonic fermented JPP (FJPP) in a 3D model of colorectal cancer (CRC) composed by HT29 spheroids. JPP samples fermented with human feces during 0, 2, 8, 24 or 48 h were incubated (10,000 µg mL^−1^) with spheroids, and cell viability was assessed after 72 h. Chemometric analyses (cluster and principal component analyses) were used to identify the main compounds responsible for the bioactive effect. The antiproliferative effect of FJPP in the CRC 3D model was increased between 8 h and 24 h of incubation, and this effect was associated with HHDP-digalloylglucose isomer and dihydroxyphenyl-γ-valerolactone. At 48 h of fermentation, the antiproliferative effect of FJPP was negligible, indicating that the presence of urolithins did not improve the bioactivity of JPP. These findings provide relevant knowledge on the role of colonic microbiota fermentation to generate active phenolic metabolites from JPP with positive impact on CRC.

## 1. Introduction

Colorectal cancer (CRC) is the third most common type of cancer and the worldwide incidence in the year 2018 was nearly two million people [1]. Almost 75% of all sporadic cases of CRC are directly affected by dietary intake, and the consumption of plant phytochemicals has been reported to be beneficial in CRC management [2]. Phytochemicals, such as phenolic compounds (PC), have chemopreventive property mostly due to their capacity to inactivate reactive oxygen species, which plays a vital role in the initiation and progression of CRC [2]. Besides, PC may also modulate cell gene expression, apoptosis or differentiation [3]. Studies regarding CRC are usually carried out using monolayer cells. In vitro 3D cell culture models using human and patient-derived CRC cell lines have been described as to be promising tools to evaluate the role of many antitumoral compounds, since these models provide cell–cell and cell–matrix interactions and, in this way, a cellular context similar to the cancer microenvironment. In particular, spheroid cultures of human CRC cells produced in stirred culture systems have been shown to be enriched in cancer stem cells and better recapitulate the in vivo tumor behavior [4,5]. Of note, these models have been used to ensure an accurate evaluation of the anticancer potential of phytochemicals derived from brassicas, nuts and citrus fruits [4,6,7,8]. 

Jaboticaba fruit (*Myrciaria trunciflora*) is a Brazilian berry with great potential for the food industry, being consumed naturally or as juices, jams, wines and liqueurs [9]. This fruit is rich in PC, in particular anthocyanins, flavonols, and ellagitannins, which are mostly concentrated in the peel [10]. Following jaboticaba processing for juice production, the peel is discarded and becomes a large food-waste. Thus, studies regarding the bioactivity of jaboticaba peel compounds will contribute to the sustainability of this agri-food system, supporting waste reuse as a nutraceutical or pharma chemical source. Accordingly, some PC extracted from jaboticaba exhibited antiproliferative effects against HT29 and HCT116 colon cell lines [11]. However, PC from jaboticaba peel powder (JPP) have low intestinal bio-accessibility and undergo extensive catabolism by gut microbiota [12]. 

The biological implication of the gut-derived PC metabolites of JPP remains unknown as previous studies on JPP fermentation were limited to the description of PC transformation during simulated digestion [12]. Although the antiproliferative effect of JPP extract in HT29 monolayer cells has been recently reported [13], no studies were found regarding the antiproliferative effects in CRC 3D cell models or the effects of colonic fermented JPP (FJPP). 

This study aimed to evaluate the antiproliferative effect of FJPP in a 3D model of CRC (HT29 spheroids). Moreover, PC composition data of FJPP were processed by multivariate analyses, such as cluster analysis (CA) and principal component analysis, to determine the relationship between PC generated during fermentation and the antiproliferative effect of FJPP. 

## 2. Results and Discussion 

The inhibition of colon cancer cell line growth by PC has gained attention as having potential for a candidate compound for cancer therapeutics [3]. However, some caution in data interpretation is needed, since PC have low bioavailability. Given that a large portion of PC are eliminated in feces, the transformation of PC by gut microbiota may influence the therapeutic potential of these compounds [14]. Besides being substrates for colonic microbiota, PC fermentation may also generate products that benefit the intestinal environment [15]. 

PC from JPP are poorly absorbed in the small intestine, and most of these compounds reach the colon and suffer fermentation by gut microbiota [12]. After simulated salivary, gastric and intestinal digestion, the fraction of JPP that was not bio-accessible (JPP-IN) was used for the colonic fermentation assay. In vitro colonic fermentation is a simple model to simulate the catabolism of compounds by colonic microbiota, and several reports have demonstrated the catabolism of PC during colonic fermentation using this model [16,17].

In vitro gut fermentation assays are relatively simple and fast procedures that present an unmatched opportunity for performing studies frequently challenged in humans and animals owing to ethical concerns. Fresh feces are the usual source of gut microbiota, but the large inter-individual variability of the gut microbiota poses a great challenge for biological replications [18]. This issue can be partially overcome by pooling fecal samples from different donors, as conducted in the present study. 

### 2.1. Antiproliferative Effect of FJPP in Monolayer Cultures and Spheroids of HT29 Cell Line

All antiproliferative assays were carried out using a feces control group, i.e., a fecal suspension alone (no JPP-IN) that was fermented for the same time as JPP, to eliminate the interference of PC already present in the feces. Before antiproliferative assays, the cytotoxicity of FJPP was tested in confluent Caco-2 cells at concentrations ranging from 125 to 10,000 µg mL^−1^. None of the evaluated concentrations caused toxicity to these cells after exposure for 72 h (Appendix A), indicating their safety for intestinal epithelial cells. The antiproliferative effect of FJPP (125 to 2000 µg mL^−1^) was firstly screened in monolayer cultures of HT29 cells, and all samples inhibited cell proliferation with EC_50_ values ranging from 769 to >2000 µg mL^−1^ (Figure 1). However, at the end of fermentation, (24 h and 48 h) the antiproliferative effect decreased when compared with time 0 (non-fermented samples) (Figure 1, *p* < 0.05).

After confirming the antiproliferative potential of FJPP in HT29 monolayer cultures, we investigated whether FJPP samples were able to impair cell proliferation in a more complex model. The HT29 spheroids cultured for seven days in a stirred culture system were selected for the study because they display characteristics observed in in vivo carcinomas, such as the hypoxic regions, the apoptotic/necrotic core, less differentiated cells in the surrounding area and a higher percentage of cancer stem cells, which has been associated with chemotherapeutic resistance [19]. 

The antiproliferative effect of FJPP was evaluated in HT29 spheroids by analyzing cell viability after 72 h of incubation with 10,000 µg mL^−1^. Time-course colonic fermentation of JPP resulted in a bell-shaped antiproliferative profile that was best described by a quadratic equation (Figure 2, Panel A, *p* < 0.05). The maximum point obtained from the first derivative showed that peak cell growth inhibition (69.6%) would occur for the FJPP obtained after 22.4 h of colonic fermentation (Figure 2, Panel A). This result is in agreement with a previous report [20], where colonic metabolites were detected in urine and plasma of subjects after 24 h of pomegranate juice intake, suggesting that this time would be enough for colon bacteria to metabolize the juice PC. After 48 h of fermentation, no significant antiproliferative effect of JPP was observed when compared with the control fermentation medium (Figure 2, Panel A, *p* > 0.05). Although the antiproliferative effect of dried peel powder and freeze-dried extract of *Myrciaria jaboticaba* fruits in HT29 cells has been recently reported [13], our study is the first to show the antiproliferative effect of JPP in a highly complex CRC model. Moreover, our study evaluated digested and colonic-fermented samples, which resembles the in vivo transformations of JPP and its effects in the colon. 

Higher PC concentration was required to inhibit spheroids proliferation compared to monolayer cells (Figure 1 vs. Figure 2, Panel A). In agreement, the antiproliferative effect of baru nuts (*Dipteryx alata* Vog) extracts was lower in HT29 spheroids when compared with the antiproliferative effect in HT29 monolayer cells [4]. This confirms a higher resistance of HT29 spheroids when compared with usual HT29 monolayer cells, most likely due to the difficulty of PC in diffusing through the cell spheroids and/or the chemo-resistant phenotype of this model, as described previously for other PC [6,8]. Interestingly, the antiproliferative profile of JPP during time-course colonic fermentation was different in HT29 monolayer cells (Figure 1) and HT29 spheroids (Figure 2, Panel A). While the highest antiproliferative effect against monolayer cells were observed for non-fermented JPP and up to 8 h fermentation, spheroids were mostly inhibited by JPP fermented during 8 h and 24 h. The high structural complexity of PC before JPP fermentation (see Section 2.2) likely posed a greater difficulty for their diffusion into HT29 spheroids, and limited their antiproliferative properties compared to the monolayer model. This way, 3D spheroids could represent a better model to evaluate the antiproliferative effect of PC from JPP when compared with HT29 monolayer cells. 

### 2.2. Chemometric Analyses to Identify Bioactive PC Metabolites

The detailed catabolism of PC during this colonic fermentation assay of digested JPP was reported in our previous study [12] and is summarized in Figure 2, Panel B. The present study brings novel information to understand how fermentation affects the antiproliferative properties of JPP and the putative PC and metabolites associated to the antiproliferative properties.

The relationship among PC, their metabolites and the antiproliferative activity during the colonic fermentation of JPP was investigated using multivariate analysis. 

#### 2.2.1. Treatment Grouping

Cluster analysis (CA) of JPP submitted to different times of colonic fermentation revealed that samples were divided into four groups (0, 2, 8/24 and 48 h of fermentation) based on their PC and metabolite composition along with the antiproliferative effect. This grouping explained 87% of data variation (Figure 3, Panel A, red dashed line). JPP fermented for 8 h and 24 h were within the same group, indicating that they have analog characteristics concerning PC composition and antiproliferative activity (Figure 3, Panel A). Principal component analysis was also used to investigate the relation among JPP from different fermentation times according to their PC composition and antiproliferative activity. The model used was constituted by three principal components that explained 91.3% of data variance (Figure 3, Panel B) and confirmed the findings of CA, as it revealed a close relationship between JPP from 8 h and 24 h of fermentation. These fermentation times were located close to each other and within the negative-central region of axis 2 and the negative regions of axes 1 and 3 (Figure 3, Panel B). 

#### 2.2.2. Variable Grouping

CA of the dependent variables (PC compounds, PC metabolites and antiproliferative activity in CRC 3D cell model) allowed PC compounds to be clustered into nine groups according to their content during the different times of colonic fermentation. This clustering explained 68.5% of data variation (Figure 4, red dashed line): Groups 1 to 9 (top to bottom) contained 1, 9, 4, 2, 3, 5, 5, 5 and 7 compounds, respectively (Figure 4). Principal component analysis allowed the visualization of PC from JPP during colonic fermentation, in an n-dimensional space, by identifying the directions in which most of the information was retained. In this analysis, the biplot can show inter-unit distances among the units as well as display variances and correlations of the variables. As far as we know, this is the first study using chemometric analyses to investigate bioactive PC and metabolites in a CRC 3D model.

The antiproliferative activity (cell growth inhibition) induced by 10,000 µg mL^−1^ in HT29 cell spheroids was clustered along with hexahydroxydiphenic (HHDP)-digalloylglucose isomer + dihydroxyphenyl-γ-valerolactone (Figure 4). A biplot (fermentation time vs. PC and metabolites content, Figure 5) presented encyclical behavior counterclockwise and confirmed the findings of CA. Furthermore, this biplot increased the proportion of variance explained to 91.3% using the first three principal components (Figure 5). The loadings of principal components are displayed in Appendix A. 

At the beginning of fermentation (0 h), the predominant PC compounds were high-molecular-weight compounds naturally found in JPP [9,11], namely hydrolysable tannins, such as penta, tetra, tri and digalloylglucose isomers and HHDP derivatives, besides anthocyanins (Figure 2, Panel B). These complex compounds were degraded over time, increasing the concentration of simpler hydrolysable tannins, such as HHDP-digalloylglucose isomer, and proanthocyanidin metabolites such as dihydroxyphenyl-γ-valerolactone, which were more closely related to the antiproliferative effect of FJPP as shown by their proximity with this bioactivity marker both in CA (Figure 4) and in principal component analysis (Figure 5). 

The biplot showed that the antiproliferative effect was closely related to 8 h and 24 h of fermentation and to HHDP-digalloylglucose isomer and dihydroxyphenyl-γ-valerolactone (Figure 5). The concentration of these compounds was transiently increased during the first stages of colonic fermentation, being thereafter decreased and not detected after 48 h of fermentation [12]. Because of the complex structure of hydrolysable tannins, they are gradually depolymerized during colonic fermentation, resulting in a transient increase in the concentration of smaller tannin polymers, such as HHDP-digalloylglucose isomer, which was already present in the undigested JPP [10]. Ellagitannins have HHDP group(s), which release ellagic acid upon hydrolysis. However, HHDP, a dimer of gallic acid from pomegranate juice, exhibited a higher antiproliferative effect by inducing necrosis in HT29 cells when compared with gallic and ellagic acids [21]. Our results suggest that hydrolysable tannins released in the colon upon consumption of JPP could potentially curtail the risk of CRC development, as previously reported for pomegranate juice [21]. Ellagitannins from *Myrciaria jaboticaba* seeds have been recently shown to exhibit chemopreventive properties against CRC by reducing inflammation and increasing proapoptotic pathways in rats [22]. 

Dihydroxyphenyl-γ-valerolactone, which was likely formed by microbial degradation of catechin and epicatechin from JPP [10], was the other compound associated to the antiproliferative activity of FJPP (Figure 4 and Figure 5). This metabolite has been shown to contribute to the urinary antioxidant activity in rats treated with (−)-epicatechin [23]. Moreover, it has been recently shown that dihydroxyphenyl-γ-valerolactone is able to reach the brain, supporting the neuroprotective effects of PC-rich foods [24]. In fact, the plasma concentrations of dihydroxyphenyl-γ-valerolactone were positively correlated with memory improvement in mice with Alzheimer disease supplemented with polyphenolic extract from blueberries and grapes for four months [25]. As far as we know, this is the first report about the bioactivity of dihydroxyphenyl-γ-valerolactone in a cancer model.

Although HHDP-digalloylglucose isomer and dihydroxyphenyl-γ-valerolactone seem to be the major compounds associated with the increased antiproliferative activity of JPP during the intermediate stages of colonic fermentation, other PC and metabolites likely contributed to the antiproliferative effect that was already found before starting colonic fermentation and after 48 h of colonic fermentation (30% and 20% cell growth inhibition; Figure 2, Panel A). 

Gut microbiota and PC have shown a synergistic effect regarding their chemopreventive properties in monolayer cell models [26], which was likely related to microorganism enzymes that convert PC into more bioavailable or bioactive forms than their parent compounds. Moreover, PC can inhibit pathogenic bacteria and favor the development of beneficial microbiota. In fact, a reduction in pathogenic bacteria count was observed concomitant with the JPP PC catabolism by human fecal microbiota [12]. Although microbiome composition has not been evaluated in the present study, the intake of a yogurt added with lyophilized seed extract of jaboticaba has been recently shown to increase the abundance of *Bacteroidetes* and decrease the number of *Firmicutes* in a rat model of chemical-induced colon cancer [27]. Lastly, as oxidative stress may be involved in the death of probiotics, PC may delay this process due to their antioxidant properties, increasing the viability of probiotics [26]. In agreement, the addition of Bifidobacterium in an extract rich in water-soluble PC from jaboticaba enhanced antioxidant activity and antiproliferative ef-fects in monolayer cancer colon cells when compared with the group without the probiotic [26]. 

At the end of fermentation, a significant reduction in antiproliferative activity was observed along with the highest concentration of final metabolites (urolithins). Urolithins, which were linearly increased during fermentation (Figure 2, Panel B), are a class of compounds produced by the gut microbiota metabolism of ellagitannins that have been suggested as biomarkers of the intake of PC from berries, nuts and wines. Moreover, urolithin A was able to decrease colony formation of monolayer human colon cancer cells [28]. However, in the present study, the increase in the antiproliferative effect of FJPP was not related to the production of urolithins, since the antiproliferative activity of FJPP decreased by 48 h of fermentation (in monolayer and 3D cells) while the urolithins content reached their maximum content at this time. Since FJPP presented a mix of different urolithins, we cannot rule out that the interactions among these metabolites may have masked a possible antiproliferative effect. 

## 3. Materials and Methods

### 3.1. FJPP Samples

All of the samples used in this work were prepared and characterized in our previous study [12]. Briefly, the digestion of JPP was performed with *Myrciaria trunciflora* fruits collected in São Vicente do Sul city, at Rio Grande do Sul State, Brazil (SISGEN ABD 4602). The peels were separated from the pulp, freeze-dried and triturated to produce JPP. JPP (5 g sample) was subjected to a sequential static in vitro simulation of oral, gastric and duodenal digestion as previously described [10]. The fraction containing PC that were not available for absorption (JPP-IN), was separated, freeze-dried and used for the in vitro colonic fermentation assay. Each 5 g of JPP yielded 4.2 g of JPP-IN.

In vitro colonic fermentation was carried out using fresh feces from human donors (eight men and nine women aged between 20 and 53 years-old) to provide gut microbiota. The assay was the same as that already described [10]. The JPP-IN fraction was incubated in glass bottles containing 50 mL of fecal suspension. Fecal suspension without the JPP-IN was also incubated and used to correct the results by the respective controls. After incubation, glass bottles were centrifuged at 1400× *g* for 10 min, the supernatant was collected and stored at −80 °C for chromatographic analysis or freeze-dried and stored frozen before cell culture assays. For cell culture experiments, the freeze-dried fermented supernatant (FJPP) samples were reconstituted with Milli-Q water and then centrifuged at 2000× *g* for 15 min. FJPP samples were filtered using 0.22 µm filter twice to yield the sterile FJPP samples. 

### 3.2. Cell-Based Assays

#### 3.2.1. Cell Lines and Culture

Human colon cancer cell lines, HT29 and Caco-2, were obtained from American Type Culture Collection (ATCC, Manassas, VA, USA) and Deutsche Sammlung von Microorganismen und Zellkulturen (Barunshweig, Germany), respectively. Both cell lines were grown in RPMI 1640 medium (Gibco, Carlsbad, CA, USA) supplemented with 10% (*v/v*) of heat-inactivated sterile filtered Fetal Bovine Serum (FBS; Biowest, Riverside, CA, USA). For Caco-2 cells, additional supplementation was made with 1% (*v/v*) of PenStrep (Gibco, Carlsbad, CA, USA). Stock cells were maintained as monolayers in 175 cm^2^ culture flasks and incubated at 37 °C with 5% CO_2_ in a humidified atmosphere. 

#### 3.2.2. 3D Cell Culture Using a Stirred-Tank Culture System

CRC spheroids (3D cells model) were generated as previously described [19] with some modifications. Briefly, HT29 single cells were inoculated into a 100 mL spinner flask (Corning, Tewksbury, MA, USA) in a culture medium with 10% (*v/v*) FBS, accounting for a cell density of 2.5 × 10^5^ cells/mL. The spinner vessel was placed on a magnetic stirrer under 40 rpm, and cell culture was carried out in a humidified atmosphere with 5% CO_2_ at 37 °C, with an increasing stirring speed to 50 rpm and 60 rpm at the time-point of 8 h and 28 h post-inoculation, respectively. After the 4th day post-inoculation, half of the spinner flask volume was renewed daily. Experiments were performed using spheroids collected at day eight of culture with an average diameter of 500 μm. 

#### 3.2.3. Cytotoxicity Assay in Caco-2 Cells

The cytotoxicity of FJPP was assessed using confluent and undifferentiated Caco-2 cells as a model of the human intestinal epithelium, as previously described [29]. Briefly, Caco-2 cells were seeded into 96-well plates at a density of 2 × 10^4^ cells/well and allowed to grow for seven days, with medium renewal every 48 h. At day seven, cells were incubated with FJPP that was diluted in a culture medium at concentrations ranging from 125 µg to 10,000 µg of JPP-IN equivalents mL^−1^ Cells incubated with culture medium and cells incubated with control fermentation medium (fecal suspension without JPP-IN) were used as controls. Another batch of control cells were incubated with Milli-Q water (FJPP vehicle) to account for culture medium dilution effects during FJPP addition. After 72 h of incubation, cells were washed with Phosphate Buffered Saline (PBS, Sigma-Aldrich, St. Louis, MO, USA) and cell viability was assessed using the 3-(4,5-dimethylthiazol-2-yl)-2,5-diphenyltetrazolium bromide (MTT) assay [4]. Cell viability was calculated relative to the respective feces control (no JPP-IN) and relative to medium control. Six independent experiments were performed in triplicate.

#### 3.2.4. Antiproliferative Assay in HT29 Cell Monolayers

HT29 cells were seeded in 96-well plates at a density of 1 × 10^5^ cells per well. After incubation for 24 h, cells were treated with several concentrations (0-control, 125, 250, 500, 1000 and 2000 µg of JPP-IN equivalents. mL^−1^) of JPP previously fermented for 0, 2, 8, 24 and 48 h (FJPP). Cells incubated with culture medium and cells incubated with control fermentation medium (fecal suspension without JPP-IN) were used as controls. After 72 h of incubation, cells were washed with PBS and cell viability was assessed using the MTT assay, as described by [4]. Cell viability was calculated relative to the respective feces control (no JPP-IN) and relative to the medium control. Six independent experiments were performed in triplicate.

#### 3.2.5. Antiproliferative Assay in HT29 Cell Spheroids (3D Cells)

HT 29 spheroids were seeded at a density of approximately five spheroids/well, in 96-well plates and incubated with PrestoBlue Viability Reagent (Molecular Probes, Invitrogen, CA, USA) to determine the basal viability. Then, spheroids were treated with FJPP at 10,000 µg of JPP-IN equivalents. mL^−1^ (five times the highest concentration used in the 2D cells antiproliferative assay, but still non-toxic). After 72 h of incubation, the spheroids were washed with PBS and cell viability was assessed using the MTT assay, as described above. Cell viability was calculated relative to the respective feces control (no JPP-IN) as previously described [8]. Six independent experiments were performed in triplicate.

### 3.3. HPLC-Q-TOF-MS/MS Analysis of PC during Colonic Fermentation

The extraction, identification and quantification of individual PC and metabolites from the FJPP of this assay were described in our previous study [12]. Briefly, anthocyanins were separated in a C-18 Core-Shell Kinetex column (2.6 μm particle size, 100 mm, 4.6 mm; Phenomenex, Torrance, CA, USA) at 38 °C using a gradient of 3% formic acid in water and 100% acetonitrile at a flow rate of 0.9 mL·min^−1^. Non-anthocyanin PC and metabolites were separated in a C-18 Hypersil Gold column (5 μm particle size, 150 mm, 4.6 mm; Thermo Fisher Scientific, Waltham, MA, USA) at 38 °C using a gradient of 5% methanol in acidified water (0.1%, *v/v*, of formic acid) and 0.1% acetonitrile at a flow rate of 1.0 mL·min^−1^. The identification of PC and metabolites was performed in a HPLC system connected to a diode array detector (DAD) and a Q-TOF mass spectrometer analyzer and electrospray ionization (ESI) source (micrOTOF-QIII, Bruker Daltonics, Bremen, Germany). Compounds were identified based on their elution order and the comparison of their UV to visible spectra and mass spectrometry fragmentation patterns with authentic standards and literature data. The quantification of PC and metabolites was conducted using DAD peak area data using the method previously validated [10]. Hydroxybenzoates were quantified at 280 nm as equivalents of gallic acid or protocatechuic acid, tannins were quantified at 280 nm as equivalents of gallic acid, anthocyanins were quantified at 520 nm as equivalents of cyanidin 3-glucoside, and flavonols and urolithins were quantified at 360 nm as equivalents of quercetin or myricetin. The limits of detection (LOD) and quantification (LOQ) for gallic acid, protocatechuic acid, cyanidin 3-glucoside, quercetin and myricetin were 0.012 and 0.037, 0.027 and 0.083, 0.020 and 0.068, 0.562 and 1.363, 0.166 and 0.503 mg·L^−1^. 

### 3.4. Statistical Analysis

Antiproliferative activity data were expressed as mean ± SEM. Statistical analyses were performed using GraphPad Prism 5 software (GraphPad Software, Inc., La Jolla, CA, USA). Data were submitted to one-way analysis of variance (ANOVA) and the means were compared by Tukey’s test at a 5% significance level. Additionally, antiproliferative activity data was also submitted to regression analysis. For antiproliferative assay in monolayer HT29 cells, the EC_50_ values were calculated using the former software. 

The bioactivity of PC metabolites formed during colonic fermentation of JPP was investigated using chemometric analyses. Principal Component Analysis and Cluster Analysis (CA) were used to investigate the association between HPLC-MS-fingerprinting assessment of PC (parent compounds and metabolites) and the antiproliferative activity during the colonic fermentation of JPP. Data were processed using the software SAS^®^ OnDemand for Academics (SAS Institute Inc., Cary, NC, USA).

## 4. Conclusions

This study demonstrated the antiproliferative effect of JPP fermented by colonic microbiota against CRC using a complex 3D cell model. The potential effects of JPP against CRC were increased in the intermediate times of fermentation, and were associated to HHDP-digalloylglucose isomer and dihydroxyphenyl-γ-valerolactone rather than to other colonic PC metabolites or to the PC found at highest concentrations in the undigested fruit. Studies regarding the antiproliferative effect of these isolated compounds in CRC 3D models should be carried out in the near future.

## Figures and Tables

**Figure 1 molecules-26-04469-f001:**
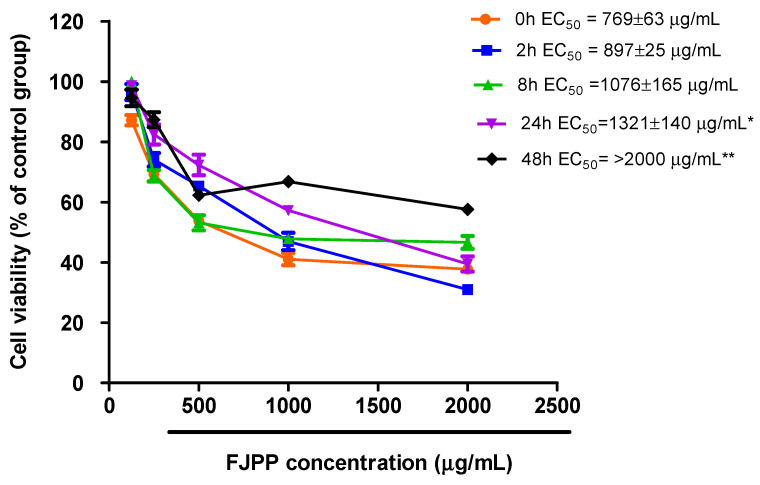
Antiproliferative effect of FJPP (0 h to 48 h of fermentation) in HT29 monolayer cells. Cell growth was evaluated after exposure to FJPP at 125 to 2000 µg mL^−1^ for 72 h. Results are means of at least 6 independent experiments performed in triplicate ± SEM. Statistical analyses were performed using GraphPad Prism 5 software (GraphPad Software, Inc., La Jolla, CA, USA). * Different from time 0. ** Different from all groups.

**Figure 2 molecules-26-04469-f002:**
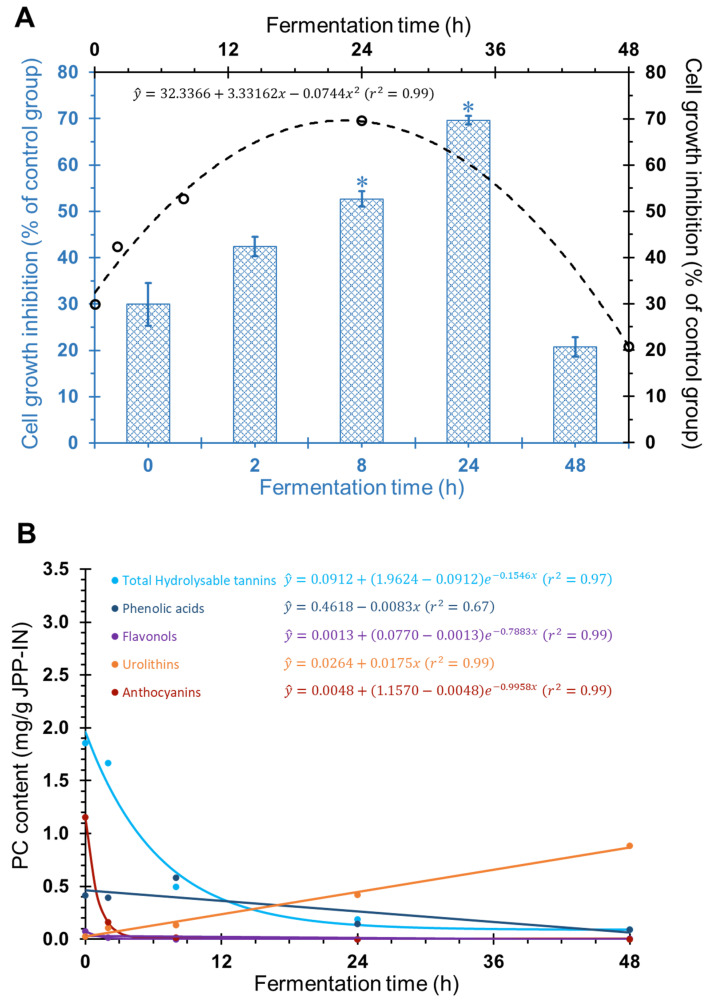
Changes in the antiproliferative effect (**A**) and in PC and metabolites content (**B**) of digested JPP (JPP-IN) during in vitro fermentation with human feces. Cell growth was evaluated after exposure of HT29 spheroids to FJPP at 10,000 µg mL^−1^ for 72 h. In panel A, data regarding the equation are displayed in black at the superior and right axes, whereas data regarding bars are displayed in blue at the bottom and left axes. Data displayed in panel B were obtained from Quatrin et al. (2020) [12]. Results are means of at least 6 independent experiments performed in triplicate ± SEM. Statistical analyses were performed using GraphPad Prism 5 software (GraphPad Software, Inc., La Jolla, CA, USA) and software SAS^®^ OnDemand for Academics (SAS Institute Inc., Cary, NC, USA). * Different from medium control group.

**Figure 3 molecules-26-04469-f003:**
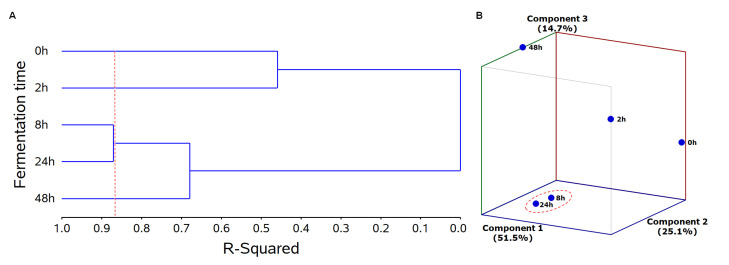
Dendrogram from CA of fermentation time (h, ordinate axis) in relation to the coefficient of determination (r2, abcissa axis) using Euclidean distance as a measure of dissimilarity and Ward’s agglomerative hierarchical algorithm as a grouping method (**A**); three-dimensional graphic dispersion of the fermentation time in relation to the main components of principal component analysis (**B**). In panel A, the red dashed line indicates the percent of data variation explained by CA, and shows the significant groups formed. Statistical analyses were performed using software SAS^®^ OnDemand for Academics (SAS Institute Inc., Cary, NC, USA).

**Figure 4 molecules-26-04469-f004:**
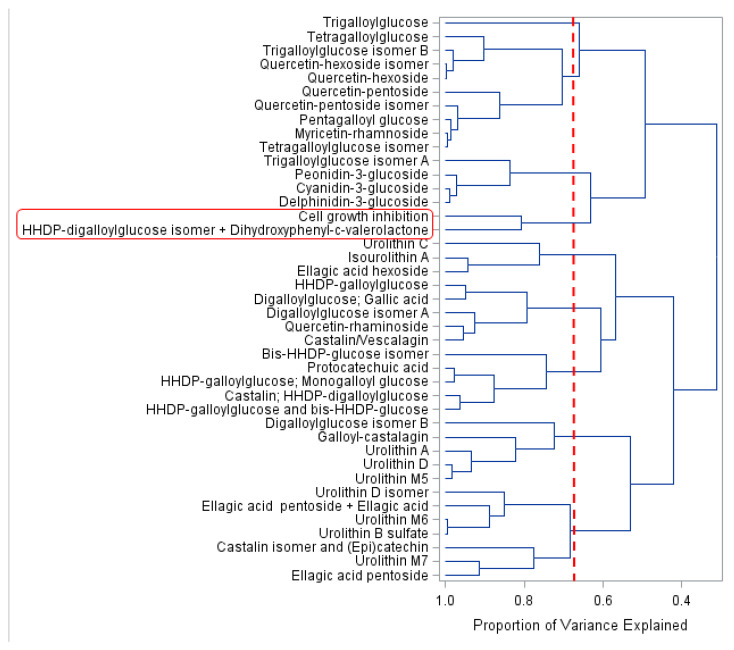
Dendrogram of PC content (mg compound/g equivalent to undigested JPP) and antiproliferative activity (%, ordinate axis) in relation to the coefficient of determination (r2, abcissa axis) using the correlation matrix as a measure of similarity and the main component as a grouping method. The red dashed line indicates the percent of data variation explained by CA and shows the significant groups formed. Antiproliferative activity in the CRC 3D cell model (cell growth inhibition) was highlighted using the red rectangle. Statistical analyses were performed using software SAS^®^ OnDemand for Academics (SAS Institute Inc., Cary, NC, USA).

**Figure 5 molecules-26-04469-f005:**
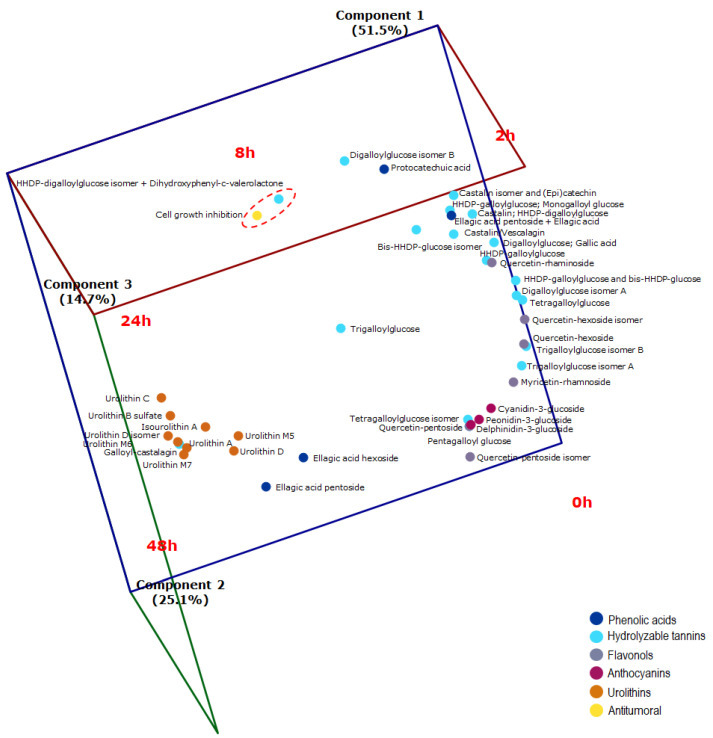
Three-dimensional biplot of fermentation time (scores) vs. PC, metabolites and antiproliferative activity (loadings) in relation to the main components of principal component analysis. A red dashed line indicates the proximity between HHDP-digalloylglucose isomer and dihydroxyphenyl-γ-valerolactone with the antiproliferative effect of FJPP. Statistical analyses were performed using software SAS^®^ OnDemand for Academics (SAS Institute Inc., Cary, NC, USA).

## Data Availability

The data supporting the findings of this study are available on request from the corresponding author. Supporting information is provided in the Appendix A.

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
