# Peer review of "Antiproliferative Effect of Colonic Fermented Phenolic Compounds from Jaboticaba (Myrciaria trunciflora) Fruit Peel in a 3D Cell Model of Colorectal Cancer"

_molecules, 2021, doi:10.3390/molecules26154469_

Round 1

Reviewer 1 Report

The manuscript entitled “Colonic fermentation of jaboticaba (Myrciaria trunciflora) fruit peel increases its antiproliferative effect in a 3D cell model of colorectal cancer a chemometric approach to identify active phenolic metabolites”, authored by Paula R. Augusti and colleagues, deals with the evaluation of the antiproliferative effects of jaboticaba peel powder after colonic fermentation.

Given the limited data, the manuscript fits with what can be considered a short communication. In general the work is well organized, although some points could be better discussed. For example, the manuscript contains several typos, such as words not written in italics (for example, in vivo), units of measure wrongly reported (for example, cm2; 2 x 104 cell/well; mg.mL-1), etc. Authors should carefully check the manuscript and correct these errors.

The figures have very low quality, and many of them are illegible. In particular:

  • The panels of figure 1 should be positioned horizontally. The letter designating the panels should be shown at the top left, black- coloured and without the square brackets.
  • Authors should colour the histograms shown in Figure 1, Panel A equally. In fact, even if the timing is different, the measurements refer to the same parameter.
  • It is not possible to understand what is reported in the writings of Figures 2 and 3.
  • The panels of Figure 2 and 3 should also be arranged horizontally.
  • Furthermore, the name of the statistical software used for the realization of these graphs should be clarified in the caption of the figures.

In the text, the reference to the figures should be reported as follows: Figure 1, Panel A, etc.

The introduction is well written and organized. However:

  • Line 47 - 48 should be reformulated. Authors should express the concept using in an impersonal way.
  • Regarding Jaboticaba, recent studies have highlighted the nutraceutical and nutritional properties of this particular fruit (https://doi.org/10.1016/j.foodchem.2019.125515). In particular, the authors should first underline the importance of bioactive components of the whole fruit, reporting the main bioactive compounds mainly identified in the pulp or peel, and only subsequently highlight that most of these are present in the peel. This topic is very important as, following the processing of food processing for the production of juices, the skins are discarded and they become a large and dangerous food-waste. The study carried out by the authors perfectly fits in a perspective of sustainability and enhancement of food waste. Authors should better address this topic as it would greatly increase the value of the manuscript.

Section 3.4. of materials and methods should be largely improved. Although the authors cited a previous work in which probably the same instrumentation and the same chromatographic and analytical conditions were employed, the type of instrumentation, column, chromatographic conditions, LOD, LOQ, quantification methodology should be clearly reported also in this manuscript. Moreover, authors should add how they identified the compounds, the detector coupled to HPLC, etc.

Data availability information is missing.

Reviewer 2 Report

Augusti et al., in their manuscript “Colonic fermentation of jaboticaba (Myrciaria trunciflora) fruit peel increases its antiproliferative effect in a 3D cell model of colorectal cancer: a chemometric approach to identify active phenolic metabolites” evaluated the antiproliferative effect of colonic fermented phenolic compounds from jaboticaba fruit peel in 3D cells of colorectal cancer.

  1. Previous manuscript by Quatrin et al., 2020 (10.1016/j.jff.2019.103714) also did the same work (Colonic fermentation of jaboticaba, HPLC analysis), antiproliferative effect is only added in the present study.
  2. Title is not informative with grammatical mistake.
  3. Provide data for FJPP effects in HT29 cell line.
  4. The present study uses human fresh feces for the colonic fermentation rather than using gut microbiota alone, what was the significance of this study. How it could be translated to in vivo and clinical studies?
  5. Figure 1A & B are same, only difference is column and line type of chart. Figure 1A & B is not complete and lack of information.
  6. Present study claims colonic fermentation by gut microbiota plays a major role. If so, gut microbiome analysis should be done.

Round 2

Reviewer 2 Report

Authors revised the manuscript according to the comments. Manuscript can be accepted for publication.